# Fundamental Pedagogical Aspects for the Implementation of Models-Based Practice in Physical Education

**DOI:** 10.3390/ijerph18137152

**Published:** 2021-07-04

**Authors:** Alejandra Hernando-Garijo, David Hortigüela-Alcalá, Pedro Antonio Sánchez-Miguel, Sixto González-Víllora

**Affiliations:** 1Department of Specific Didactics, Faculty of Education, University of Burgos, 09001 Burgos, Spain; ahgarijo@ubu.es; 2Department of Didactics of Musical, Plastic and Body Expression, Teacher Training College, University of Extremadura, 10003 Cáceres, Spain; pesanchezm@unex.es; 3Department of Physical Education, Arts Education, and Music, Faculty of Education, Cuenca, University of Castilla-La Mancha, 16071 Ciudad Real, Spain; sixto.gonzalez@uclm.es

**Keywords:** physical education, pedagogical models, models-based practice, learning, involvement

## Abstract

The implementation of pedagogical models (PMs) in the subject of Physical Education (PE) is presented as a pedagogical approach that is based on the educational context as a means to overcome the serious limitations that arise from traditional approaches. The effective implementation of this approach has demonstrated benefits in terms of student motivation, student involvement and improved learning. Thus, its application and international relevance, the variability of content covered, the possibility of replicability in a variety of contexts and the fact that it favors a reflective framework and common action by teachers are some of the reasons that justify its use. In this sense, the need for teacher training, as well as the intention to generate more scientific evidence based on its application in the classroom, are some of the key aspects to be taken into account for its implementation and consequent consolidation in the educational field.

## 1. Introduction

PE is the subject with the greatest impact on the motor development of children and young people. Additionally, it is also one of the most relevant areas for the social and psycho-evolutionary development of individuals [1]. In fact, PE is not only concerned with students’ physical aptitudes and motor skills, but it also offers them unique opportunities for the development of their cognitive and affective capacities and the development of a meaningful understanding of the importance of incorporating PE into their lives. Motor skills in this subject is the tool that acts as the means for the achievement of learning in all spheres that completes the holistic development of students [2,3,4,5,6].

The generation of successful experiences and the impact they have on students will determine the use of PE, its transferability and its subsequent applicability [7,8,9]. Furthermore, the interaction of variables such as the creation of a motivational climate, students’ perceived competence and their achievement goals in PE, demonstrate the choice of physical activity outside the school context in their leisure time [10].

Adherence, understood as the voluntary choice of students to take part in physical activity and sport outside school hours, is one of the priority aims of the subject in order to compensate and complement the time devoted to the subject of PE at school [10].

To achieve such purposes, it is necessary to question the most effective way of delivering PE, although in this respect, there does not seem to be widespread agreement and consensus [11]. Boonsem and Chaoensupmanee (2020) [12] determine the factors of effectiveness of teaching in PE. Among them, methodology is one of the essential elements that will give meaning and validity to teaching practice [12,13].

However, the choice of methodology must be based on scientific evidence that is directed towards the true aims of PE and gives it rigor as a subject [14]. In this way, the insertion of pedagogical models (PMs) and their incorporation into the classroom is increasingly taking shape with scientific backing and support [15]. They try to distance themselves from the more traditional methods that have prevailed in the subject and whose validity is questioned from the point of view of motivation, involvement and learning on the part of students [16].

Some authors point out that teaching models have evolved from a perspective that considered the teacher as the central axis from which the teaching process was articulated (teacher-centered approach), to another that emphasizes the role of the student in a context in which responsibility and autonomy are encouraged (student-centered approach) [13]. This second perspective, in which the teacher is the guide and facilitator of student learning, is the basis for the functioning of the PMs. These constitute a worldwide methodological approach in the field of PE. Some of them are called “basic” due to their consolidation in the field of education and their research history (Cooperative Learning, Sports Education, Comprehensive and Personal and Social Responsibility) and others are considered “emerging” due to their incipient and progressive establishment (Adventure Education, Motor Literacy, Attitudinal Style, Ludotechnical, Self-Construction of Materials, Health Education) [17,18].

Cooperative Learning is a pedagogical model in which students learn with, from and by other students through an educational approach that facilitates and enhances this positive interaction and interdependence. For a learning structure to be considered cooperative, it has to fulfil five essential elements: (1) Positive Interdependence, (2) Promoting Interaction, (3) Individual Responsibility, (4) Group Processing and (5) Social Skills.

The Sport Education model aims to create authentic sport experiences for students by experiencing sport in a holistic way and developing their motor competence and sport culture. 

The Comprehensive model, or Teaching Games for Understanding, starts from the modified game of sport so that the student understands the structure of the sport, its tactics and the necessary technical skills, in that order. 

The Personal and Social Responsibility model finds in the practice of sport the means to work on respect and positive relationships with others. To this end, six levels of responsibility are proposed in a gradual sequence: Level 0: irresponsible behavior and attitudes; Level 1: respect for the rights and feelings of others; Level 2: participation and effort; Level 3: autonomy and leadership; Level 4: help; Level 5: transfer.

In terms of emerging models, the Attitudinal Style states that generating a positive attitude in learners is the origin for better learning and increased motivation towards PE. It aims for all learners, regardless of their physical abilities, to have positive experiences. It is developed through three fundamental elements: (1) the Intentional Body Activities, (2) the Sequential Organization towards Attitudes; and (3) the Final Assemblies. Among the emerging models, this one is selected because it is established as one of the most transversal models in terms of its applicability, because it allows a clear interdisciplinarity with other areas, because it seeks the positive attitude of the student towards the tasks and because it is articulated around these three easily replicable elements that can be used by teachers to implement it in the classroom in a basic way [19].

These models have a number of specific characteristics and therefore, pursue specific objectives [19]. However, they share common elements that underpin their practical application and even allow them to be combined and hybridized [20]. Common aspects include the fact that they all start from a theoretical basis that determines their structure and implementation, the consideration of the student’s maturity, the teacher’s own expectations and experiences, the creation of favorable learning environments and the importance of the organization and evaluation process.

PMs have meant a transition towards a new pedagogical conception and even a redefinition of the purposes of PE [21]. They are even presented as a means to re-question the “status quo” of PE and to verify whether its establishment really meets the needs that were missing with the application of other more traditional approaches [22].

Any pedagogical change must be studied to ensure its true and sufficient potential or, on the contrary, whether it is an approach that needs to be reconsidered [23]. Despite their long history, it is still necessary to investigate whether these models are being effectively introduced in education [24]. Therefore, generating scientific evidence from the classroom provides a framework for further teaching application.

Models-based practices in PE have been developed in recent years as a way for teachers and students to structure the pedagogical approach according to the subject matter and learning context [25]. These are presented as an opportunity to renew and improve the way of teaching in PE by generating real evidence of learning and achievement [26]. 

The aim of this article is to justify the implementation of PM in PE, as well as to establish the key aspects for its implementation in the classroom.

In this way, it should be made clear that this is not an experimental study, but rather it is a theoretical foundation based on the most current and relevant literature on the subject. The databases on which this content review has been structured (Tylor and Francis, Eric, Scopus, Journal Citation Reports, Scimago Journal Rank) are international in scope and the references included are mainly taken from the last 10 years.

## 2. Why Implement Models-Based Practice in PE?

Models-based practice in PE focuses on the strong interaction between student, teacher, content and context in the teaching–learning process [17,27]. 

However, the interdependent use of these elements must be preceded by teacher training and updating that demonstrates the mastery and control of basic educational parameters such as curriculum development and its adequacy in the use of resources to meet all relevant issues in the implementation of its design. Sometimes, this is not favored by initial training, in-service training or the role of the Education Administration itself in terms of the scarcity of such teacher training, among other factors. In this sense, it is important to consider that the establishment of these basic pillars must be a precursor to the implementation of PM.

Thus, the justification for the implementation of PM is determined by several reasons such as: (a) they are internationally relevant, (b) they deal with a variety of contents and focus on the teacher and the student, (c) they enable students to be involved in what they do, favoring their self-regulation and motivation, (d) they allow a diversity of applications thanks to the possibility of hybridization, (e) they guarantee replicability in a diversity of contexts, (f) they encourage all teachers to act under a common framework, and (g) they encourage joint reflection as a professional group [20]. 

Models-based practices in PE have international scientific relevance as they have been tested in a variety of contexts through the analysis of multiple variables, demonstrating how they work for student learning [15,18]. In fact, the need to build a rigorous evidence base to justify their inclusion as an essential component of PE is emphasized [28]. Pozo, Grao-Cruces and Pérez-Ordás (2018) [29] point out that the analysis of the relevance of the model focuses on the study of variables such as the following: (a) the impact of the teaching program based on the application of the model itself; (b) the implementation characteristics of the program; and (c) the outcomes of the program on the participating students. In this sense, when the starting point is evidence of success, the teacher feels more confident, and its implementation shared by the group lends rigor to the subject of PE [23]. 

Models-based practice address a diversity of content; in fact, they cover the complexity and variety of the existing curricular structure [30]. However, their implementation is teacher- and learner-centered, taking into account the context and resources available beyond the choice of the specific content [31]. Thus, the model-based approach requires an investment in which teachers give students a more central role in the classroom by encouraging their responsibility and self-regulation in order to facilitate transfer outside the classroom [20,32].

Pedagogical practices based on models have been shown to generate a higher level of autonomy and motivation on the part of students as they enable their involvement in what they do, favoring their satisfaction and perception of achievement [13]. Calderón et al. (2013) [13] studied the perception of students in the application of basketball and hockey under the comparison of the Sports Education model with the Traditional model. Two fifth grade classes were involved, both of which were taught by the same specialist teacher. The 5th A class consisted of 17 students and the 5th B class consisted of 16 students. A simple quasi-experimental randomized crossover design with non-probabilistic convenience sampling was used, in which the two levels of the independent variable (teaching models) were applied to the two participating classes. After data analysis, it was observed that the perception of learning, satisfaction, enjoyment and involvement in the tasks was perceived as motivating when practicing under the premises of Sport Education, and especially in the group that experienced it after the Traditional model.

The perception of motivation is linked to learning outcomes [20]. Precisely, the interaction of these variables is related to adherence to physical activity, which is one of the most important aims of the subject [10]. Models such as Cooperative Learning or the Attitudinal Style focus their development on the involvement of the learner as a fundamental element in the teaching and learning process [33]. For example, Cooperative Learning is implemented on the basis of five basic components: positive interdependence; promoting interaction; individual responsibility; group processing; and social skills [34], which are based on the need for active student involvement from the beginning. For its part, the Attitudinal Style finds in the achievement of skills by all, regardless of their personal characteristics, the need for learners to get involved and help each other to achieve it [33].

In addition, the model-based approach allows for a diversity of applications due to its hybridization. Hybridization involves combining the most significant elements of one or more models either by extracting and merging the key features of each model or by taking one of the models as fundamental and including techniques, strategies or resources from other models [35]. There are several proposals on the hybridization and combination of different models, where it is pointed out that the combination is more beneficial for student learning than if one model is applied in isolation [17]. In studies such as those of González-Víllora et al. (2019) [35], it is found that the hybridization of sports models such as Sports Education and Teaching Games for Understanding (TGfU) favors the improvement of skills related to the game, referring both to technical–tactical aspects and to the understanding of the game. Hybrid models that include Cooperative Learning and/or Teaching for Personal and Social Responsibility favor the improvement of psychosocial, personal and affective variables.

The adoption of a model-based approach conceives the teaching and learning process as a dynamic system, made up of dynamic variables (student, context and task) that interact with each other, and whose correct development shapes quality learning [13]. Therefore, replicability in different contexts is guaranteed, because these contexts pose a challenge that often leads the teacher to adapt the application of the model by demonstrating a modified version of PE that manifests itself in practice [30,36].

The ideal application of pedagogical models is framed within the different educational stages that make up the school context. Its scope of application has been extended to the primary, secondary and initial teacher training stages [8,10,13,17]. With regard to the latter educational stage, Pérez-Pueyo et al. (2020) [17] analyzed the perception of prospective teachers on the usefulness and transferability of the Attitudinal Style model in their classes. Twelve prospective PE teachers from the University of Burgos (Spain), aged 20.14 years, participated. A qualitative approach was used with two data collection instruments (focus group diaries and discussion group) and two categories of analysis were established: (a) usefulness in the construction of professional identity; (b) transferability of the Attitudinal Style in the school. The results showed how the future teachers considered the model as a transcendental methodological tool for approaching PE at school. On the other hand, they identified interpersonal relationships, learner autonomy and group responsibility as fundamental characteristics of the model and with high transferability at school.

However, the framework of application goes beyond the school context by establishing the link and liaison with the out-of-school environment. This is especially true in the example of the Sport Education model, in which some of its structural elements are season, affiliation or final event. Schwamberger and Sinelnikov (2015) tried to connect what students experience in PE with their participation in physical activity outside school through the Sport Education model. A class of 50 fifth graders participated in a 20-session football program. The weekly program sessions included (1) a session held during EF class at school, and (2) a session held after school at the local park. The results of the program were positive in connecting in-school curricular instruction with out-of-school student participation.

Similarly, the application of models extrapolates their application and consequent replicability to other educational contexts. As an example, models such as Personal and Social Responsibility or Cooperative Learning have demonstrated their transversality when implemented in educational areas other than PE [7,10]. Turgut and Gülşen Turgut (2018) examined the effects of Cooperative Learning on mathematics achievement in Turkey using the meta-analysis method, identifying that the model had a positive influence on this variable.

Ultimately, all teachers are encouraged to operate under a common framework. The model-based approach involves pedagogical and curricular branching, a process of discontinuity through which new interactions, new practices and new learning are organized [23,37,38]. This can become a pedagogical dilemma, but it is recognized that the main benefits and/or successes attributed to its use are related to such dilemmas [30]. Therefore, teacher training and pedagogical knowledge of the approach are fundamental to the implementation of quality PE [24]. This approach gives rise to continuous joint reflection as a collective, favoring critical awareness of one’s own practice and allowing the exchange of ideas on new pedagogical actions that establish a reconstructed notion of the educational value of PE [16,39].

## 3. Key Aspects for Implementation in the Classroom

On the interdependence of relationships among content, context and teaching–learning, several key aspects that a teacher should consider to implement PM in the classroom are revealed: 1. show clarity about what is taught and learned in the classroom; 2. incorporate them into their professional identity; 3. select content as a means to achieve learning; 4. link the use of models to motivational aspects of the learner; 5. encourage learner self-regulation and autonomy; 6. allow for the hybridization of models; 7. integrate evaluation in a purposeful way.

### 3.1. Teacher Clarity about What Is Being Taught and Learned in the Classroom

Model-based approaches to PE have been developed in recent years as a way for teachers and students to focus on a manageable number of learning objectives and to align the methodological approach with the subject matter and learning context [25]. In any case, they are based on pedagogical principles aiming at meaningful PE [40]. However, model-based practices are no substitute for critical questioning that even redefines the purposes of the subject [21].

Reflectively and systematically considering what students should to learn in PE entails the need to be clear about what we want to teach in the classroom [21,26,41]. Therefore, research designs must include how teachers give value to what they teach and, consequently, how students attach meaning to what they learn [32].

For example, Cooperative Learning should not have as its ultimate goal that students learn to cooperate but that they cooperate to learn [42] in order to acquire the basics of physical fitness, to learn a sporting and/or expressive modality or to educate through direct, experiential learning in and through physical activities in the natural environment [15,43,44,45,46].

### 3.2. Professional Identity of the Physical Education Teacher

Professional identity is a shaping element of the teaching profession and of the teacher himself/herself. It is constructed and developed and is therefore constantly being renewed and reconfigured [36,37,38]. The implementation of models involves pedagogical change, and for this to be fully established, it must be part of the educational conception and professional identity of the PE teacher [27]. It therefore involves understanding the epistemological complexities and pedagogical implications despite the difficulties teachers may encounter in not progressing as quickly as they would like to change their practice in line with their philosophy [2,20,27].

This process of identity readjustment is strongly influenced by the following practices: (a) teacher training; (b) the need for long-term implementation; and (c) collaboration from supportive agencies [20,23,32,47].

Adequate teacher training in this field is an important tool in the development of a reflective professional identity [39]. Therefore, when teachers have a clear conception and master the variables that regulate PM, learning outcomes in the different domains of PE are higher [14,20,23,24].

However, Hastie and Wallhead (2016) [32] indicate the need for sufficient time to consider changes in philosophy and practice. This could be achieved through long, slow but gradual action-based research designs that allow close monitoring of the implementation of these pedagogical approaches.

Similarly, Casey (2014) [23] considers that the most important factor in bringing about change is school–university collaboration. These supportive relationships allow teachers to continuously reconsider their practice with the help of experienced peers who contrast and evidence results from a scientific point of view [20].

### 3.3. Choice of Content as a Means of Learning Achievement

The choice of a model is not associated with the teaching of a given content. In fact, the same content can be approached from different methodological perspectives and thus from the application of a variety of PMs. Thus, the model-based approach allows choice, and works on disparate content [18].

However, the implementation of PMs goes beyond the content to be worked on. For example, the Sport Education model was designed so that all students, regardless of their skills and abilities, would have successful experiences in learning sport [48]. For its part, the application of the TGfU model leads to the best tactical understanding and practice of the sport being applied [49]. Additionally, the Attitudinal Style model demonstrates that by taking as a basis the choice of any content (jumping rope, juggling, acrobatics, fencing, football, shadow theatre, orienteering, etc.), group achievements are obtained by all the pupils as a result of collaborative and cooperative work [33].

### 3.4. Linking with the Motivational Aspects of the Learner

The effective implementation of PMs in classrooms increases student motivation and even their emotional intelligence, enhancing the development of more equitable and inclusive classroom environments [32,50]. In order to do so, the PE approach based on PMs must be linked to the interests of the learner. In addition, providing students with achievement goals that are processed and facilitated on a frequent basis will increase their awareness of their own successes and achievements [50]. This will support the potential transfer of sport education experiences to physical activity settings beyond PE [32].

For example, the Attitudinal Style is a model that precisely finds in the motivational aspect the fundamental initial gear so that the physical activity proposed generates autonomy to be practiced autonomously both inside and outside the classroom [33]. Other models such as Adventure Education, which are developed around a playful and action-based environment, have been shown to have a positive impact on students’ overall self-esteem as well as on their social skills and motivation towards tasks [44,51,52]. Similarly, models such as Cooperative Learning or Personal and Social Responsibility are linked to motivational aspects of the student, as they involve them from the beginning in their teaching–learning process [33,51].

### 3.5. Encouraging Self-Regulation and Student Autonomy

The implementation of models is associated with the promotion of self-regulation and consequent learner autonomy. Therefore, their application, combined with the integration of formative assessment processes, can have an impact on the improvement of students’ emancipation and learning management [53,54].

The inherent characteristics of some models allow for the promotion of such autonomy and independence. For example, the Personal and Social Responsibility Model is based on empowering students to develop positive skills, values and behaviors through sport so that these can be transferred outside the school context [4,7,8,9]. However, in order for its implementation to be successful, it will be necessary to implement guidelines for individual and group responsibility in a clear and progressive manner, depending on the level of autonomy that students acquire in the selected content [17,18,55]. Similarly, it is necessary to include assessment processes that allow the learner to be aware of the level of responsibility acquired [53].

Other models, such as TGfU or Sports Education, have been shown to work intentionally on the assumption of responsibility and decision making on the part of the student, improving their autonomy. For this method to be successful, it is essential that the teacher guarantees the students’ active involvement on decisions about the game itself and its structure: dimensions of the field, number of participants, playing time, use of material or restrictions in relation to contact [56].

### 3.6. Possibility of Hybridizing Models

PMs have their own characteristics depending on their objectives and fields of application. In this sense, hybridization means that the student simultaneously experiences contrasting methodologies that at the same time have common characteristics that guide and facilitate their connection [35,49].

Therefore, another aspect to be taken into account by teachers in the implementation of PMs is that they should be able to allow hybridization in order to achieve success: (a) overcome the limitations of the application of single and isolated PMs; (b) generate a diversity of applications of the contents; (c) carry out a more individualized education adapted to each specific context; (d) promote better results in the different learning domains (motor, cognitive, social and affective) [19,27,35,49,57].

Based on these proven benefits, there are a multitude of proposals for hybridizing the different models, some of which have shown evidence of success in student learning [57]. For example, the combination of Cooperative Learning and TGfU can improve students’ responsibility, skill level and understanding of sport within a positive working environment [15,58,59,60]. Other models that are frequently hybridized due to the common characteristics they share are TGfU and Sport Education [56]. Their hybridization results in authentic learning experiences [57]. Similarly, the Social and Personal Responsibility model has been hybridized with others, such as Sport Education or Cooperative Learning, showing a positive impact on student behavior and learning [51,61,62].

Similarly, the Social and Personal Responsibility model has been hybridized with others, such as Sport Education or Cooperative Learning, showing a positive impact on student behavior and learning [51,61,62]. Other models frequently hybridized due to the common characteristics they share are TGfU and Sport Education [56]. Their hybridization results in authentic learning experiences [57].

In this sense, Farias, Mesquita and Hastie [57] analyze the impact of a football teaching unit with the hybridization of the Sport Education model and the Comprehensive model. The study was conducted in Portugal; it involved 24 students with an average age of 10.3 years and lasted 17 sessions. The Blomqvist, Vänttinen and Luhtanen (2005) instrument was used to assess game performance as well as game comprehension in terms of decision making and skill execution in 10 min matches. It was concluded that the combined application of both models allows for an authentic learning environment in which learning tasks focused on specific football skills are developed. This implementation promoted in students’ performance improvements and understanding of the game, and increased correlations between the two constructs were also observed.

### 3.7. Integration of the Evaluation

The application of PMs makes real sense when an evaluation process is associated with it [59]. Assessment is an integral part of the teaching–learning process, which provides valuable information to both students and teachers. Thus, assessment intentionally integrated into the different models facilitates student learning while allowing teachers to determine what students need along the way [63].

In this sense, formative and shared assessment can be a key tool for the optimal and effective implementation of PMs to generate learning [8,54].

The fact that students acquire the ability to self-evaluate and co-evaluate their practice means that the teacher hands over responsibility to the students, and thus, that they are connected to the content and to their own improvement process [64]. In this respect, it is also necessary that the procedures and instruments used are accessible and simple to understand for student use. On the other hand, it is essential that the assessment be authentic. For example, in the application of a Sport Unit carried out through the TGfU model, authentic assessment means that students are assessed during the actual practice of the sport being developed and not in isolated and decontextualized tasks. In this way, the student will understand the essence of the sport they are playing and can use that knowledge outside of the school context [54,59,64,65,66,67].

## 4. Reflections and Future Lines of Work

Teaching intervention based on the use of teaching models makes it possible to propose a well-structured and quality teaching and learning process in PE. In this sense, PMs are emerging strongly as a means of overcoming the major limitations that the application of traditional approaches has shown in terms of educational benefits for students [13]. The reductionist (mechanistic, technical vision with little consideration for diversified teaching) has prevailed in the way the subject has been developed [26]. Therefore, model-based practice offers a possible solution to these problems by designing appropriate teaching strategies, making concrete and working towards pre-established learning outcomes [16].

Although much progress has been made around this new approach to PE, work is still being carried out to ensure that it takes hold, bearing in mind the need for time to consider real changes in philosophy and the consequent implementation of successful practice by teachers [58]. To this end, realistic challenges need to be set that should arise from training, experience and effort to engage in the planning and dissemination of a broader notion of PE linked to the implementation of PMs [20].

In this sense, the main contributions of this work are focused on the importance of the use of the models, the need for a more flexible curriculum that allows the hybridization of the models and the necessary teacher training. These aspects constitute fundamental bases for the educational community of PE teachers to implement the PMs in the classroom.

On the basis of the contributions presented, future work streams, therefore, focus on achieving aspirations of educational value with a long-term scope. These are related to the need to (a) have a clear vision of the aims of the subject of PE; (b) conceive motor skills as a means of learning and not so much as an end in itself; (c) connect with the interests of students by enhancing their abilities and competences, and, ultimately; (d) generate more and more scientific evidence from the classroom to validate the theoretical premises defined from each of the models. In this sense, there should be a transfer between what is learned in PE classes and the lifestyle of the students and, therefore, students should be active during non-school hours.

Being aware that “not everything goes in PE” allows for an important exercise of reflection on the awareness that teachers must have and transmit to students about the clear reasons that the subject contributes to the integral education of people. It is important to understand that current PE has three main purposes as a specific curricular subject within the education system: (1) the physical–motor development of pupils; (2) the creation and recreation of pupils’ physical culture; and (3) its contribution to the global approach to pupils’ all-round development [41].

This approach means conceiving that motor skills will always be present, but often, more than as an end, as a means to achieve other domains that make up the personality of the individual [35]. In this respect, López-Pastor et al. (2016) [57] point out that the application of models and above all, their hybridization, makes it possible to start from motor skills to stimulate the cognitive structuring of pupils, providing explanatory structures that organize and give meaning to learning. In addition, the hybridization of pedagogical models allows for better adaptation and individualization to the specific practice context of each group of learners. On the other hand, it requires the teacher to have more in-depth training in the pedagogical models he/she intends to hybridize.

On the other hand, a reflective and systematic approach to what we want our students to learn in PE is a priority. To this end, the teacher must pay special attention to what students learn and in which direction this learning is taking place [12,26]. In this sense, it is argued that a model-based approach, together with a reconstructed notion of the educational value of PE, can offer a possible future for the subject [16]. Its application takes place in authentic learning environments in which students participate, decide, are involved, understand the interrelationships of physical sport activities and are able to transfer knowledge to other contexts [57].

In any case, there is a need to build a rigorous evidence base to support their inclusion as an essential component of the subject matter and to further explore their educational impact on student learning [13,28,66].

In this sense, the use of cyclical action research processes can be an alternative to generate knowledge about teaching and learning and increase and improve understanding in practice [67,68,69]. This is in spite of the existing gap between academic discourse and classroom reality, as teachers themselves often consider research as an activity that has little to do with their daily practice [58]. In this regard, Gubacs-Collins (2011) [67] points out that the use of action research gives a voice to students and teachers, who are the protagonists of action in the framework of the application of these approaches. Casey and Dyson [58] used action research as a framework to investigate cooperative learning and tactical games in a tennis unit with 800 11–12-year-olds. The results of this research reinforce the concept that the implementation of any new pedagogical approach requires a lot of time and work [70,71]. However, it highlights the conceptual and practical change made by the teacher/researcher to give autonomy to the students in their learning process. 

The investigation of PM from the classroom implementation itself can provide an empirical insight, and an extraction of generalizable results on the use of an approach involving one or several models to teach PE [20]. This investigation will make it possible to study and establish new perspectives on this renewed pedagogical approach. However, it is highlighted to conceive them not as recipes but as pedagogical reference points in the interest of improving working methods and quality educational practices that aim for better learning and greater satisfaction with practice on the part of pupils and teachers [25,72,73].

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
