# Peer review of "Fundamental Pedagogical Aspects for the Implementation of Models-Based Practice in Physical Education"

_ijerph, 2021, doi:10.3390/ijerph18137152_

Round 1

Reviewer 1 Report

Dear authors, the text, after the corrections made, is good. I still believe that the study should refer to the territorial range of the analyzed literature, the period of the analyzed research, as well as the databases that were taken into account, but it is not necessary. Listing at the end of at least three conclusions related to the assumed goal would increase the value of your work.

Author Response

Thank you very much for your review. We resubmit the manuscript in response to your comments. 

Reviewer 2 Report

While the authors did make some changes, this piece still is not focused enough for publication.  The curricular models, now discussed, need to be tightly interwoven into the manuscript.  What is the importance of those model as related to this paper, other models are mentioned as secondary, but why pick the Attitudinal Style.  The real focus of this study, is why and if the selected models are used and their effectiveness on the context.  An interesting premise, however, the review still isn't focused. Teachers are not trained in curriculum development, backward design, appropriateness of grade level, gender appropriateness and resource assessment in order to meet all relevant issues in application of curriculum design, and yet the author(s) do not discuss this.  This paper, needs to be reconsidered in terms of focus and scope, before it is a candidate for publication.  Finally, the revisions are poorly written.

Author Response

(The authors gave the same response as above.)

Reviewer 3 Report

IJERPH-1252197

First, off I would like to thank the authors for their great work and contribution to physical education. I think it is important for teachers to grasp the different physical education models and understand the purpose and alignment to the lessons' intent. Too often, I see teachers implement a particular educational model only having a surface-level understanding. This provided no benefits to the students. Having a clear understanding of the origin of the model and how the model aligns with state and national physical education standards is vital. If one aims to teach the whole child they must select a model that allows for all three domains of learning and growth and/or a model that can be combined with another without losing the essence of its purpose. I commend the authors for their work and how it was laid out. For the authors, I do not have any major concerns regarding the content in your article. I think it was well done; however, I would offer some suggestions that might help with the grammar and flow of your paper. Thank you.

Line 30: motor skills, but it also offers them unique

Line 36: transferability and its subsequent

Line 38: goals in PE, demonstrate

Line 57: (teacher-centered approach), to another

Line 87: fundamental elements: (1)

Line 88: towards attitudes; and (3)

Line 99: application of other (remove comma) more traditional

Line 102: whether these models are being effectively…

Line 111-112: study, but rather it is a theoretical foundation…

Line 122: common framework, and g)

Line 129: variables such as the following: (a)

Line 131: program instead of programme

Line 134: diversity of content; in fact,

Line 136: resources available (remove comma) beyond

Line 159-160: five basic components: ...interdependence; …interaction; …responsibility; …processing;

Line 175: favor instead favour

Line 186: analyzed instead of analysed

Line 191: were established: (a)

Line 220: This approach gives rise to…

Line 221: favoring instead of favouring

Line 250: himself/herself.

Line 252: change; and for this action to be fully established

Line 257: influence by the following practices: a)

Line 281: [49]. However, the Attitudinal Style Model

Line 321: For this, For this method to be successful, it is essential…

Line 331: order to achieve success:

Line 352: in Portugal; it involved 24

Line 357: This implementation promoted in

Line 358: of the game and as well as increased

Line 383-384: reductionist (mechanistic, technical vision with little considerations for diversified teaching) has

Line 393: Future work streams, therefore, focus

Line 400: students (remove comma) and, therefore,

Line 403: [41] It is important to understand

Line 408: This approach means…

Line 411: models (remove comma) and, above all their…

Line 430: practice. [67] This is despite Despite the existing gap

Line 431: reality, as teachers themselves often consider research

Line 443: This investigation will make it possible

Author Response

Thank you very much for your review. We resubmit the manuscript in response to your comments. 

This manuscript is a resubmission of an earlier submission. The following is a list of the peer review reports and author responses from that submission.

Round 1

Reviewer 1 Report

The review article submitted for review is very general. It lacks a thorough analysis of the topic.
The article lacks several important elements such as: the type of the analysed literature base, which period it concerned, which age group, which country or countries it referred to. 
There was also no information on how physical education classes are run nowadays (subject range, hours, basic objectives), where sports classes are carried out besides school and what pedagogical models are used there.
The importance of physical education classes in the development of children and young people was not mentioned clearly enough.  The article should have discussed in detail the pedagogical models used and recommended. In chapters 2 and 3 there are no references to specific results, only general final statements of the research results are quoted.

Reviewer 2 Report

While this paper has an interesting premise, the pedagogical focus linked with curricular models, the paper is not presented in a comprehensive way, and there is no method of how the review was conducted.  In addition, the models that are discussed are not described, and the pedagogical methods presented for each model, is not discussed fully, but in a general manner.  This paper is not detailed enough to consider for publication.